# Hypoxia and Oxygen-Sensing Signaling in Gene Regulation and Cancer Progression

**DOI:** 10.3390/ijms21218162

**Published:** 2020-10-31

**Authors:** Guang Yang, Rachel Shi, Qing Zhang

**Affiliations:** Department of Pathology, University of Texas Southwestern Medical Center, Dallas, TX 75390, USA; 1120160389@mail.nankai.edu.cn (G.Y.); rachel.shi@utsouthwestern.edu (R.S.)

**Keywords:** hypoxia, PHDs, TETs, JmjCs, HIFs

## Abstract

Oxygen homeostasis regulation is the most fundamental cellular process for adjusting physiological oxygen variations, and its irregularity leads to various human diseases, including cancer. Hypoxia is closely associated with cancer development, and hypoxia/oxygen-sensing signaling plays critical roles in the modulation of cancer progression. The key molecules of the hypoxia/oxygen-sensing signaling include the transcriptional regulator hypoxia-inducible factor (HIF) which widely controls oxygen responsive genes, the central members of the 2-oxoglutarate (2-OG)-dependent dioxygenases, such as prolyl hydroxylase (PHD or EglN), and an E3 ubiquitin ligase component for HIF degeneration called von Hippel–Lindau (encoding protein pVHL). In this review, we summarize the current knowledge about the canonical hypoxia signaling, HIF transcription factors, and pVHL. In addition, the role of 2-OG-dependent enzymes, such as DNA/RNA-modifying enzymes, JmjC domain-containing enzymes, and prolyl hydroxylases, in gene regulation of cancer progression, is specifically reviewed. We also discuss the therapeutic advancement of targeting hypoxia and oxygen sensing pathways in cancer.

## 1. Introduction

Molecular oxygen serves as a co-factor in many biochemical processes and is fundamental for aerobic organisms to maintain intracellular ATP levels [1,2]. Responses to low oxygen or hypoxia are highly conserved and play critical roles in pathological and physiological processes [3,4]. Oxygen distribution gradients are complicated, and oxygen levels are extremely heterogeneous among various tissues [5]. Hypoxia, a condition of insufficient oxygen levels to maintain regular cellular function, is not defined by a fixed oxygen concentration, since some tissues work at 5% oxygen equivalent to an atmosphere and others as low as 1% oxygen [6]. Mechanisms that control oxygen homeostasis and the subsequent modulation of metabolism and development work over an extensive range of oxygen concentrations and in response to transient challenges that span from seconds to days, weeks, or months [6]. A breakthrough in our understanding of cellular responses to oxygen changes is the identification of hypoxia-inducible factors (HIFs), and their regulation by the von Hippel–Lindau (VHL) tumor suppressor protein (pVHL) and prolyl hydroxylases (PHD1, PHD2, PHD3 or called EglN2, EglN1, EGLN3, respectively) of the 2-oxoglutarate (or α-ketoglutarate) dioxygenase superfamily [6]. The interactions of these molecules form a fundamental molecular framework for how variations of oxygen levels can affect transcriptional responses and provides therapeutic targets for multiple diseases, such as anemia, cardiovascular disease, and cancer [1]. This groundbreaking research was justifiably honored with the 2019 Nobel Prize in Physiology or Medicine awarded jointly to William G. Kaelin Jr, Sir Peter J. Ratcliffe, and Gregg L. Semenza ‘for their discoveries of how cells sense and adapt to oxygen availability’ [7].

Cancer remains a global public health problem with high mortality, poor prognosis, and few effective treatment choices [8]. Around 90% of cancers form solid tumors that not only distort and destroy normal tissues but also produce specific lesion microenvironments [9]. The tumor microenvironment (TME) has been well-recognized as a principal cultivator of malignancies such as invasiveness, neovascularization, and drug resistance from adaptive gene expression [10]. Uncontrolled proliferation of tumor cells reduces oxygen availability; as a result, hypoxia, or insufficient blood supply, is a typical characteristic in almost all solid TMEs [11,12,13,14]. Sub-regions of hypoxia can occur from obstructed oxygen supply, in which irregular and disordered tumor vasculature decreases oxygen availability [15]. Hypoxia can also be generated by variations in oxygen demand, and altered tumor metabolism can enhance the intracellular need for oxygen, potentially expanding hypoxia signaling to liquid tumors [16,17]. To facilitate growth under hypoxic conditions, tumors upregulate angiogenic signaling through the expression of vascular endothelial growth factor (VEGF) which increases tumor vascularization [11].

The adaptation of tumor cells to this imbalance in oxygen demand and supply is correlated with poor clinical outcomes in multiple cancer models, attributed at least in part to hypoxia-related clonal selection and genomic instability [18,19,20,21,22]. Hypoxia influences changes in gene expression and subsequent proteomic alterations that alter cellular and physiological functions, eventually worsening patient prognosis [22]. In addition, hypoxia generates intratumoral oxygen gradients, contributing to the increased expression of HIFs and the heterogeneity, plasticity, and metastatic phenotype of tumors [22]. Together, hypoxia and HIFs are known to reprogram cancer cells by regulating the expression of many genes that control angiogenesis, metabolism, cancer cell invasion, and metastasis [23,24,25]. Other core regulators in oxygen sensing signaling, including pVHL and 2-oxoglutarate-dependent enzymes, have also been identified to play an essential role in gene expression and protein homeostasis during cancer development in both HIFs-dependent and -independent manners [26,27,28].

In this review, we discuss the current knowledge about the canonical hypoxia signaling, including the function of oxygen-sensing HIF transcription factors, prolyl hydroxylation, and pVHL. The role of 2-oxoglutarate (2-OG)-dependent enzymes, such as DNA/RNA-modifying enzymes, JmjC domain-containing enzymes, and prolyl hydroxylases, in gene regulation of cancer progression, is also concisely reviewed. Finally, we summarize the therapeutic implications of targeting hypoxia and oxygen sensing pathways in cancer.

## 2. Canonical Hypoxia Signaling

### 2.1. HIF Transcription Factors: The Central Regulator of Oxygen Homeostasis

The capacity to recognize and adapt to variations in oxygen is meaningful for cellular and whole-organism homeostasis and survival. Nearly all mammalian cells respond to reduced oxygen availability by controlling the transcription factor HIF [29]. Of the 3 encoded HIF isoforms in humans (i.e., HIF-1, HIF-2, and HIF-3), the transcriptional responses to HIF-1 and HIF-2 isoforms, specifically, have been the most studied.

HIF-1 is a heterodimer of basic-helix-loop-helix-Per-ARNT-Sim proteins-HIF-1α (encoded by HIF1A) and HIF-1β (encoded by aryl hydrocarbon receptor nuclear translocator or ARNT) [30]. HIF-1α mediates the HIF complex’s oxygen sensitivity and interacts with HIF-1β by an authorized Per–ARNT–Sim (PAS) domain [6]. Both HIF-1α and HIF-1β comprise an N-terminal basic-helix-loop-helix domain responsible for DNA binding and C-terminal transactivation domains which induce gene expression [6]. The HIF-2α and HIF-3α complexes with HIF-1β are known as HIF-2 and HIF-3 respectively. HIF-α proteins are oxygen-sensitive in that they contain an oxygen-dependent degradation (ODD) domain with target prolyl residues whose hydroxylation mediates interaction with pVHL, and also the C-terminal transactivation domain, which contains the target asparaginyl residue [6,31,32] (Figure 1). When oxygen is present, hydroxylation of the prolyl or asparaginyl residue by a prolyl hydroxylase or factor-inhibiting HIF (FIH) results in pVHL binding and subsequent HIF proteasome-dependent degradation. Under hypoxic conditions, pVHL-mediated degradation does not occur, and HIF-1α stabilization allows the generation of the HIF-1 heterodimer, which is translocated to the nucleus and occupies hypoxia-responsive elements (HREs)-containing gene promoters and their transcriptional enhancers [6]. Human HIF-1 has been identified as a protein complex binding to a regulatory DNA sequence at the *EPO* locus [33]. Cooperation with other DNA-binding proteins can coactivate HIF, fine-tuning HIF targets [34,35].

HIF-1 and HIF-2 both transduce positive transcriptional responses to hypoxia, although their transcriptional targets, the kinetics of activation, and oxygen dependence differ [6,36,37]. These variations might partly be attributed to different transcriptional feedback circles. For example, an antisense HIF transcript might negatively modulate HIF-1α expression by destabilizing HIF-1α mRNA [38]. Despite binding to an identical DNA sequence at HREs, HIF-1 and HIF-2 may direct largely distinct transcriptional systems [39,40,41,42]. For instance, many metabolic responses are dependent on HIF-1, whereas cell differentiation, reparative pathways, and more complex adaptive responses to hypoxia, including induction of erythropoiesis, are dependent on HIF-2 [43,44,45]. Interestingly, HIF-1 and HIF-2 also have different patterns of genomic distribution concerning their transcriptional targets. While HIF-1 generally binds to DNA at sites in close proximity to target gene promoters, HIF-2 frequently binds to transcriptional enhancers that lie at a distance from the target gene promoter [39], and this pattern is maintained across cell types with quite different complements of HIF target genes [46]. Although some HREs can be recognized by both HIF-1 and HIF-2, very little cross-competition is observed, so that a HRE can be bound by only one HIF. The two isoforms have distinct chromatin binding patterns [46], which may be connected post-DNA binding mechanisms that mediate transcriptional selectivity [47,48,49]. Together with cell-type-specific expression of HIF-α isoforms, DNA binding selectivity generates highly distinct functional outputs for HIF-1 and HIF-2.

To date, investigation of the human transcriptional targets and tissue/cell-type specificity of HIF-3 expression has been much less complete since there are several splicing isoforms of *HIF-3A* gene products [50]. Some studies report that HIF-3α might negatively regulate HIF-1α and HIF-2α [51]. For instance, a splice variant, inhibitory PAS protein domain (IPAS), competitively binds to HIF-1/2α, preventing binding to DNA and inhibiting the HIF-1-mediated transcriptional response [52,53]. On the other hand, a study on zebrafish Hif-3α, an orthologue to HIF-3, reported that overexpressed Hif-3α directs an extensive positive transcriptional response [54]. Furthermore, of more than 150 genes that were found to be upregulated by Hif-3α in zebrafish embryos, almost 100 were also Hif-1 targets. These experiments also suggest that Hif-3α might preferentially target specific hypoxia pathways, such as the Janus kinase (JAK)–STAT and NOD-like receptor (NLR) signaling, and that at least some of the targets of zebrafish Hif-3 might also be responsive to HIF-3 in human cells [54].

Genes controlled directly or indirectly by HIF mediate an extensive spectrum of biological outputs. At the cellular level, responses to HIF include effects on differentiation, migration, cytoprotection, apoptosis, cycle control, and mitochondrial function [55,56,57]. At the organ or organism level, coordinated responses to altered HIF activity include metabolic regulation, erythropoiesis, angiogenesis, tissue remodeling, and wound healing [58,59,60,61]. The aforementioned transcriptional adaptations enable normal cells or tissues to survive in a severely oxygen-deficient environment. On the other hand, in the context of cancer, these processes can facilitate tumor progression [62]. Distinctly, in solid tumors, certain regions usually undergo hypoxic status due to abnormal blood vessel development [23]. Accordingly, this was accompanied by elevated HIF-α levels in many solid tumors. HIF activation remodels cellular metabolic oxidative mechanisms, enabling cells to mitigate toxic reactive oxygen species and preserve the synthesis of macromolecules in response to oxygen availability. Hypoxia and accumulation of HIF is also involved in resistance to chemotherapy and radiotherapy and worse prognosis in cancer patients [62]. The HIF-mediated reprogramming of varied and numerous cellular processes such as metabolism, proliferation, angiogenesis, metastasis, invasion, epithelial–mesenchymal transition (EMT), anti-apoptosis, growth factor signaling, and stem cell maintenance [62,63,64,65,66,67,68,69], underscores the significant role of HIF in cancer progression and tumorigeneses.

### 2.2. Prolyl Hydroxylation: The Adaptive Mediator of HIFs

Identification of HIF-α proteins as the oxygen-dependent subunits of HIF prompted subsequent efforts to define the molecular mechanisms of oxygen sensing upstream of the HIF signal transduction pathway [6]. As a first step, domains within the HIF-α polypeptide that could confer oxygen-regulated instability onto heterologous proteins were characterized [70,71]. Surprisingly, these domains did not seem to be regulated by protein phosphorylation pathways, which had been widely predicted to transduce the oxygen-sensitive signal [72].

In the presence of oxygen, HIF-α is trans-4-hydroxylated at prolyl residues (Pro402 and Pro564 in HIF-1α; Pro405 and Pro531 in HIF-2α; Pro492 in HIF-3α) [73,74], resulting in a >1000-fold increase in affinity for pVHL. Further analyses identified pVHL as the recognition component of an E3 ubiquitin ligase complex that targets HIF-α for proteasomal destruction by binding to the domains of HIF-α that confer oxygen-regulated instability [75,76]. pVHL is required for the oxygen-dependent proteolysis of HIF-α [77]. Post-translational hydroxylation of specific amino acid residues in HIF-α is the critical oxygen-regulated event promoting HIF-pVHL association [73,74]. Mechanistically, prolyl hydroxylation of HIF-α is coupled to the oxidative decarboxylation of prolyl hydroxylase creating a ferryl intermediate at the catalytic site [6]. HIF prolyl hydroxylation and degradation are highly efficient, so that HIF is very labile and barely detectable in normoxia. As oxygen level decreases, prolyl hydroxylation diminishes, thus enabling HIF levels to rise and the hypoxic response to be “switched on.” Prolyl hydroxylases are discussed more extensively in Section 3.1.

### 2.3. pVHL: The Proteolysis Modulator of HIFs

The physiological and pathological function of pVHL has constantly been explored since the first description of VHL disease in the early 1990s by William G. Kaelin Jr. and colleagues [78]. VHL syndrome is an autosomal-dominant, hereditary neoplastic disease associated with clear cell renal cell carcinoma (ccRCC), central nervous system hemangioblastomas, and pheochromocytoma [79]. The disease is caused by germline mutations in *VHL*, a tumor-suppressor gene located on chromosome 3p25.1 [80]. Patients who inherit a single faulty copy of *VHL* develop the disease only after spontaneous inactivation or loss of the second, wild-type *VHL* allele. The leading cause of death in patients with VHL disease is ccRCC [79]. The product of *VHL*, pVHL, is a 30 kDa protein with multiple functions. The best-documented of these functions relates to its role as the substrate-recognition component of an E3 ubiquitin ligase complex that also contains elongin B, elongin C, cullin-2, and RING-box protein 1 [75,78,81,82,83,84]. This complex is best known for its ability to target HIFαs for polyubiquitination and proteasomal degradation [75,85,86] (Figure 1), although it has been reported to target other substrates including atypical protein kinase C, N-Myc downstream-regulated gene 3 (NDRG3), Akt, erythropoietin receptor (EPOR), transcription factor B-Myb, actin cross-linker filamin A (FLNA), centrosomal protein 68 (CEP68), ceramide kinase-like protein (CERKL), and hsRPB7 [87,88,89,90,91]. In addition, by developing an in vitro VHL-capture-based binding assay combined with a genome-wide screening strategy, we have demonstrated that the zinc fingers and homeoboxes 2 (ZHX2) and the Scm-like with four malignant brain tumor domains 1 (SFMBT1) transcription factors served as novel pVHL substrates in ccRCC [92,93].

An increasing amount of evidence indicates the existence of HIF-independent pathways in the VHL-supervised background, as the simple inhibition of the HIF system might not be sufficient to prevent tumor progression. Therefore, the discovery of additional pVHL targets is critical for the development of new therapeutics. In ccRCC, a novel pVHL substrate, TANK binding kinase 1 (TBK1), an essential kinase involved in the innate immune response, was recently discovered [94]. We found that VHL-deficient kidney tumors display elevated TBK1 phosphorylation. Through genetic ablation, pharmacologic inhibition, or new carbon-based proteolysis targeting chimera specifically, TBK1 was depleted/inhibited in vitro, suppressing VHL-deficient kidney cancer cell proliferation while having no effect on VHL wild-type cells. Similarly, TBK1 depletion abrogates kidney tumorigenesis in an orthotopic xenograft tumor model [94]. Mechanistically, hydroxylation on Pro48 of TBK1 triggers both pVHL and phosphatase PPM1B binding, leading to TBK1 de-phosphorylation. TBK1 contributes to kidney cancer cell growth by phosphorylating p62/SQSTM1 on Ser366, thus stabilizing p62. In this way, TBK1, distinct from its innate immune signaling role, might serve as a novel target with therapeutic potential for cancers with VHL loss.

It is essential to note that pVHL targets other proteins in an E3 ubiquitin-ligase-independent manner with or without oxygen signal involvement. Proteins subjected to this regulatory fashion by pVHL are, for example, TBK1, aldehyde dehydrogenase 2 (ALDH2), p53, AKT, and Caspase Recruitment Domain Family Member 9 (Card9) [94,95,96,97,98]. In summary, these findings highlight the critical role of pVHL in the oxygen-signaling pathway and the control of the abundance or activity of its substrates in the context of disease.

## 3. 2-Oxoglutarate-Dependent Enzymes in Gene Regulation and Cancer Progression

### 3.1. Proline and Asparagine Hydroxylases

As mentioned previously, prolyl hydroxylation plays critical roles in the hypoxia-sensing mechanism of the HIF system. Prolyl hydroxylases belong to a superfamily of 2-OG-dependent dioxygenase enzymes that function in all eukaryotes and many prokaryotes but not Archaea. 2-OG enzymes include prolyl hydroxylase domain (PHD or EglN) enzymes, DNA/RNA-modifying enzymes, and JmjC domain-containing enzymes [97,98]. In humans, three PHD enzymes (PHD1, PHD2, and PHD3) which are equivalent to EGLN2, EGLN1, and EGLN3, respectively, catalyze C-4 prolyl hydroxylation in the N- and C-terminal oxygen-dependent degradation domains (NODDs and CODDs) of HIF isoforms [99]. Prolyl hydroxylation of HIFαs increases affinity to pVHL which tags the protein for degradation.

The activity of known prolyl hydroxylases is dependent on levels of oxygen, Fe^2+^, ascorbate, and 2-oxoglutarate [100,101]. There is substantial evidence from genetic and biochemical studies that the PHDs act as necessary oxygen sensors for the HIF system [102] with all three enzymes characterized by K_m_ values for O_2_ above anticipated cellular and tissue oxygen levels [103]. Although all of the PHDs regulate HIF, PHD2/EglN1 is the most widely expressed HIF prolyl hydroxylase, the primary regulator of HIF-1 activity in vivo, and a critical factor for survival [99,104]. Genetic inactivation of *Egln1* in mice leads to severe placental and developmental heart defects and is lethal in embryonic life [105]. By contrast, the roles of PHD1 (EglN2) and PHD3 (EglN3) in HIF regulation seem to be specific to cellular environments. PHD3 is induced by severe hypoxia by serving as a transcriptional target for HIF [106] and, therefore, important at lower oxygen concentrations. From an evolutionary standpoint, it makes sense that oxygen monitoring in more complex organisms such as mammals utilizes distinct PHDs that have different oxygen affinities and are capable of performing biochemical functions under differential conditions.

While all three PHDs can hydroxylate HIF in vitro [107], there are differential regulatory effects between them on HIF; PHD2 has a greater effect on HIF-1α while PHD3 can be more active in the regulation of HIF-2 than of HIF-1 [104]. In addition, HIF substrate discrimination is mediated by the chemical interactions between PHD loop and HIF-α CODD or NODD regions [108]. Specific naturally occurring mutations within this region of human PHD2 have marked effects on selectivity for the prolyl hydroxylation at the NODD or CODD [108]. As a result that PHDs are implicated in the HIF oxygen-sensing system, they have value as a therapeutic target, but current PHD inhibitors in clinical trials do not manifest selectivity for specific PHDs [108]. Nonetheless, these findings suggest that the design of selective small-molecule PHD inhibitors to target specific regions of PHD might be possible [108].

In addition to HIF, it should be noted that other essential proteins are subjected to PHD-mediated prolyl hydroxylation, such as MAPK6, NDRG3, FOXO3a, EPOR, ZHX2, ADSL, and Akt, suggesting potential involvement in a variety of diseases [92,97,109,110,111,112,113]. In addition, EglN2/PHD1 can act as a transcriptional activator by binding to PGC1α and NRF1 signaling complex in breast cancer [114]. We will not go through all the details here due to space limitations.

Another hydroxylase, FIH, modulates HIF-α by catalyzing the C-3 hydroxylation of an asparagine residue in the C-terminal transactivation domain (Asn803 in HIF-1α and Asn851 in HIF-2α). This modification substantially reduces the otherwise tight binding of HIF-α to the p300-CREB-binding protein transcriptional co-activators [115,116]. Thus, in contrast to PHD-catalyzed HIF-α hydroxylation, which induces a protein–protein interaction, that of FIH prevents a protein–protein interaction. Studies suggest that FIH inhibits HIF-1 [117] but also transcriptionally affects genes (*CXCR4* and *VEGFA*) in an HIF-independent manner [118]. Whereas PHDs belong to the same structural subfamily as the collagen prolyl hydroxylases, FIH has a sequence region with homology in jumonji (Jmj) transcription factors which are a part of the cupin metalloenzyme structural superfamily [115,119]. The distinct structures of FIH and the PHDs are reflected in their biochemical properties. Hypoxia reduces the activity of PHDs before that of FIH which continues to hydroxylate the asparagine residue of HIF-α [120,121].

In addition to HIF-α, FIH has been shown to target other substrates such as proteins from the ankyrin repeat domain (ARD) structural family, including Notch, transcription factors, ion channels, and cytoskeletal ARD proteins [122,123]. However, not all of the substrates undergo hydroxylation and other residues such as aspartate and histidine might be hydroxylated [124,125], and the physiological relevance of these alternative substrate hydroxylation has not been established. Research on the physiological effects of FIH-mediated signaling is still in its infancy, but FIH seems to have partial functions that are HIF-independent since metabolic signatures differ between FIH knockout mice and HIF-activated animals [126].

It is important to remain open to discover additional 2-OG dioxygenases that can be oncogenic or tumor suppressive, independently from their canonical functions. For instance, we have recently demonstrated that BBOX1, a hydroxylase that is associated with carnitine biosynthesis, contributes to triple-negative breast cancer tumorigenesis by binding and stabilizing inositol-1,4,5-triphosphate receptor type 3 (IP3R3) [127]. Depletion of BBOX1 genetically or pharmacologically inhibits TNBC tumor growth both in vitro and in vivo in a fashion independent from the enzyme’s canonical role in carnitine synthesis [127].

### 3.2. DNA/RNA-Modifying Enzymes

The most common epigenetic modification in DNA of mammals is the methylation of cytosine to 5-methylcytosine (5mC), and for a while, it was assumed that methylation, which was associated with gene repression, was an irreversible process. However, ten eleven translocation (TET) enzymes (TET1, TET2, and TET3) were recently discovered as iron- and 2-OG-dependent DNA-modifying enzymes that play an important epigenetic role by mediating the demethylation of DNA. Specifically, the TET enzymes hydroxylate 5mC to generate 5-hydroxymethycytosine (5hmC) which can be further oxidized into aldehyde 5-formylcytosine (5fC) and carboxylic acid 5-carboxylcytosine (5caC) [128,129,130]. The TET-mediated conversion of 5mC to 5hmC, 5fC, and 5caC results in the generation of guanine or oxidized cytosine base-pair mismatches [131]. He et al., found that 5caC and 5fC but not 5hmC could be cleaved and excised by DNA mismatch repair enzyme, thymine–DNA glycosylase (TDG) [128]. Base excision is followed by base-excision repair and replacement with an unmethylated cytosine. Others have suggested an alternative mechanism in which 5caC residues are substrates for an as yet unidentified enzyme that can decarboxylate 5caC to generate unmethylated cytosine [131]. In either case, oxidation of 5mC by TET enzymes, followed by TDG-mediated base excision or 5caC decarboxylation, results in net cytosine demethylation. Overexpression of the myeloid tumor suppressor, TET2, in HEK293 cells results in decreased 5mC levels and increased 5hmC, 5fC, and 5caC levels, whereas knockdown of TET1 in mouse embryonic stem cells decreased 5hmC, 5fC, and 5caC levels [131].

Physiologically, *Tet3* is critical for embryonic development [130], and *Tet1* and *Tet2* are important for hematopoietic cell differentiation [132]. In diseases, TET proteins have received the greatest research attention in the context of hematological malignancies such as acute myeloid leukemia (AML) and myelodysplastic syndrome [133,134], along with metabolic and epigenetic dysregulation that is a hallmark of cancer. TET activity is known to be regulated by levels of 2-OG, CXXC4, a negative regulator of Wnt signaling [135], calpain, a calcium-dependent cysteine protease [136], and O-linked β-N-acetylglucosamine transferase which potentially modulates TET stability [137]. Mutations in all three TET proteins have been identified in colorectal cancer [138], and *TET2* mutations and/or deletions are a characteristic of 16% of ccRCCs [139]. In addition, 5hmC product is reduced in many solid tumors and is correlated with TET expression and tumor growth [140,141]. Recently, Fan et al. demonstrate that TET2 and TET3 are novel targets for proteasomal degradation mediated by pVHL [142]. However due to genetic heterogeneity among solid tumors, it remains to be investigated whether TET deficiency is directly involved in tumor progression in animal models. In addition, there is still uncertainty regarding the direct link between TET proteins and global hypomethylation and localized hypermethylation of certain CpG island promoters that is a characteristic of cancer cells [143].

Whereas TET enzymes modify DNA, *N*6-Methyladenosine (m^6^A) demethylase is an RNA-modifying enzyme that demethylates m^6^A, the most common modification of mRNA in mammals. The m^6^A modification in mRNA and non-coding RNAs appears to impact stem cell differentiation, tissue development, and mRNA stability and fate [144,145]. Studies on the role of fat mass and obesity-associated protein (FTO), a 2-OG- and iron-dependent m^6^A RNA demethylase, in cancer are rapidly accumulating. It was recently reported that FTO is highly expressed in certain subtypes of AMLs, including t(11q23)/MLL-rearranged, *FLT3*-ITD, and/or *NPM1*-mutated AMLs and facilitates leukemogenesis, the proliferation of AML cells, and anti-apoptotic effects [146]. Similar effects were observed in lung squamous cell carcinoma and breast cancer [147,148]. In addition, there is evidence that m6A demethylation influences stem cell self-renewal; ALKBH5, which like FTO reduces levels of m^6^A, was found to increase the stability of NANOG mRNA resulting in an increase in NANOG mRNA and protein levels and enrichment in phenotype of breast cancer stem cells [65]. Pharmacological inhibition of FTO suppresses glioblastoma stem cell (GSC) growth and self-renewal in vitro and improves survival of GSC-grafted mice [149]. These findings emphasize an oncogenic role of FTO and ALKBH5 in cancer stem cells by regulating tumor initiation and development. However, questions remain on whether the functions of FTO and ALKBH5 overlap and whether FTO- and ALKBH5-regulated mRNAs are also regulated by HIFs.

### 3.3. JmjC Domain-containing Enzymes

DNA is wrapped around a histone octamer forming a unit known as a nucleosome, and the modifications on the core histones H2A, H2B, H3, or H4, can modulate transcription and epigenetic information. Although the methylation of histones has been known for decades, it is only in 2004 that enzymes that catalyze the reverse process, histone demethylation, have been identified and characterized. Petidylarginine deiminase 4 and flavin-dependent lysine specific demethylases (LSDs) were the first enzymes discovered with the capacity to reverse histone methylations [150,151], followed by the identification of the third and largest family, the Jumonji C (JmjC)-domain containing enzymes, which consists of JmjC-domain containing histone lysine demethylases (KDMs, JHDMs, or JMJDs) [151]. KDMs can orchestrate the demethylation of three lysine-methylation sites in the histone H3 N-terminal tail, a reaction that is depending on iron and 2-OG [151,152]. The proposed mechanism for KDMs consists of a hemiaminal intermediate, which likely spontaneously collapses to give the demethylated product and formaldehyde [153]. It is notable that the KDMs can act on trimethylated nitrogen sites and that N-methylation and demethylation can be either transcriptionally activating or repressive depending on the context.

Hypoxia is associated with increased histone methylation, and studies have shown that the demethylases, KDM6A (demethylates H3K27), JMJD3, JMJD6, and LSD1, have an affinity for oxygen, suggesting implications in oxygen sensing and hypoxic reprogramming. Chakraborty et al. and Batie et al. independently found a direct link between hypoxia and histone methylation [154,155], showing that KDM6A, KDM5A, or KDM3A can directly sense oxygen level and control cell differentiation through diverse downstream cascades. Specifically, KDM6A, but not its paralog KDM6B, utilizes oxygen to promote H3K27 demethylation but under hypoxia, this function is abolished [154]. Qian et al. also reported a link between KDM3A, oxygen availability, and mitochondrial biogenesis [156]. Mechanistically, under normoxic conditions, KDM3A binds to and demethylates proliferator-activated receptor-gamma coactivator-1 alpha (PGC-1α ) at lysine 224, increasing PGC-1α activity, an essential process for NRF1/NRF2-mediated mitochondrial biogenesis [156].

The past 15 years have seen significant advances in our understanding of the physiological and pathological consequences of 2-OG oxygenase-catalyzed alterations to nucleic acids and post-translational modifications to proteins (Figure 2). While we have discussed prolyl hydroxylases, TET DNA hydroxylases, JmjC domain histone lysine demethylases in this review, there are about 50 other understudied 2-OG-dependent dioxygenases, such as EGF-like domain hydroxylases, lysyl hydroxylases, and ribosomal oxygenases. Some proteins are known to have multifunctional roles, such as JMJD6 which can act as a histone arginine demethylase or a lysyl hydroxylase [157,158]. It remains to be investigated whether there is any relationship between the oxygen-sensing system, cancer pathology, and the understudied or other unidentified 2-OG dioxygenases. Nevertheless, new insights connecting 2-OG dioxygenases with the HIF–VHL system continue to be made and have been explored in the therapeutic context.

## 4. Therapeutic Implications on Targeting Hypoxia and Oxygen Sensing Pathways in Cancer

### 4.1. Inhibitors for HIF-1α

Pharmacological modulation of HIF signaling and activity represents a promising therapeutic approach for numerous diseases. While stimulating HIF signaling, such as through inhibiting PHD activity, may be advisable for certain diseases including myocardial infarction [160] or ischemia-induced tissue necrosis [161], the inhibition of HIFs and 2-OG-dependent enzymes is a primary objective in cancer treatment. HIF signaling is a hallmark of various malignant tumors with tumor suppressor function loss, oncogene function gain, intertumoral hypoxia, and unique profiles of cytokines, growth factors, etc. [162]. Increased HIF is associated with tumors from the skin, bladder, pancreas, brain, ovary, lung, liver, esophagus, uterus, colon, breast, and kidney [163]. HIF regulates gene expression which in turn, affects tumor invasion and metastasis, immune evasion, vascularization, metabolic reprogramming, autocrine growth factor signaling, cell immortalization, stem cell specification, and resistance to chemotherapy and radiation therapy.

Given the undeniable evidence of HIF in cancer pathology, there have been an increasing number of chemical compounds and approved agents designed to target HIF signaling by various molecular mechanisms (Table 1). Some of them can repress tumor growth, metastasis, and vascularization in different cancer models [164]. A variety of strategies have been attempted, such as reducing HIF-1α mRNA levels, restraining HIF transcriptional activity, decreasing HIF binding to DNA, repressing HIF-1α protein production, stimulating HIF-1α degradation, and blocking HIF subunit heterodimerization [164]. It has been reported that apigenin can down-regulate HIF-1 protein expression by targeting PI3K/AKT signaling in ovarian cancer cells [165]. Apigenin attenuates angiogenesis by inhibiting HIF-1 and VEGF expression in vivo [166]. SU5416, as a tyrosine kinase inhibitor targeting the VEGF receptor, blocks the expression of VEGF and HIF-1α [167]. SU5416 transcriptionally diminishes VEGF expression by inhibiting HIF-1 DNA binding activity through the decrease of p70S6K1 phosphorylation, AKT phosphorylation, and PI3K activity [168,169]. In addition, PX-478 down-regulates HIF-1 expression at multiple levels, including modulating HIF-1α deubiquitylation, inhibiting HIF-1α transcription and translation, and decreasing HIF- 1α mRNA [170]. BAY 87-2243 inhibits the accumulation of HIF-1α and HIF-2α efficiently and specifically [171]. Moracin O and moracin P extracted from *Morus* species inhibit HIF-1 activation with IC_50_ of 0.14 μmol and 0.65 μmol in Hep3B cells, respectively [172]. YC-1, as platelet aggregation and vascular contraction inhibitor via activating soluble guanylyl cyclase, efficiently induces the C-terminal HIF-1 degradation by targeting PI3K/Akt/mTOR signaling and repressing NF-κB activation during hypoxia [173]. Similarly, cardamonin inhibits the growth of a triple-negative breast cancer cell line by inhibiting the mTOR/p70S6K pathway and HIF-1α expression [174]. There are many more compounds with therapeutic potential that were not able to be recognized in this review, and some are described in another review paper [175].

### 4.2. Inhibitors for HIF-2α

HIF-2α non-specific antagonists commonly restrain either the transcription, protein synthesis, nuclear translocation, degradation, and accumulation of HIF-2α [176]. HIF-2α specific inhibitors such as PT-2385, PT-2399, and TC-S 7009 principally block HIF-2α-ARNT dimerization by binding to the HIF-2α PAS-B domain [175,176]. A few more compounds, including THS-017, THS-020, and THS-044, are able to bind with HIF-2α, though their activities are still unclear [176]. PT-2399, a 2,3-dihydro-1H-inden-4-yl-oxy derivative (Peloton Therapeutics, Inc., now Merck, Kenilworth, NJ, USA), and PT-2385 specifically inhibit HIF-2α-ARNT dimerization, leading to anti-angiogenic influences in human primary and metastatic VHL-defective ccRCC cell xenografted mouse models [175,177,178,179]. PT-2385 reduces the levels of human tumor-induced circulating VEGFA in the human lung cancer cell xenografted mouse model [180]. The administration of PT-2385 results in pharmacokinetics values of T_max_ = 2 h, C_max_ = 3.1 mg/m, and a half-life of 17 h and shows a 2% complete response, 12% partial response, and 52% stable response in a patient population with advanced or metastatic ccRCC and pretreated with VEGF inhibitors [180]. PT-2385 treatment also enhances the anti-metastatic activity of sorafenib by repressing HIF-2α-related Stat-3/Akt/Erk signaling in human hepatocellular carcinoma cells [181].

TC-S 7009, a 4-nitro-2,1,3-benzoxadiazole derivative, binds to the HIF-2α PAS-B internal cavity causing conformational changes in PAS-B β-sheets and preventing DNA binding, heterodimerization, and the transcription activation of HIF-2α [182]. Furthermore, TC-S 7009 specifically represses HIF-2α-induced EPO, but not HIF-1α-induced PGK1 in hypoxic hepatoma cells [177]. Finally, HIF-2α translation inhibitor-76 (Compound 76) is a thienylhydrazone compound that specifically reduces HIF-2α and not HIF-1α translation by activating the interaction between IRP1 and HIF-2α mRNA, and thus inhibiting erythrocytosis and abnormal vascular proliferation in VHL-deficient zebrafish embryos [183]. In addition, Compound-76 is able to repress chemo-resistance and stemness by inhibiting HIF-2α-EFEMP1 hypoxic signaling in a human breast cancer stem cell xenografted mouse model [184].

### 4.3. Inhibitors for 2-OG-Dependent Enzymes

Inhibitors for HIF hydroxylases—PHDs in particular—present significant therapeutic potential in the treatment of anemia and cancer, and numerous have been developed in recent years. Several pan-prolyl hydroxylase inhibitors are at the late stage of clinical trials for the treatment of anemia associated with chronic kidney disease, including Molidustat (BAY 85-3934, Daprodustat (GSK- 1278863), Vadadustat (AKB-6548), and Roxadustat (FG-4592) [186]. The ability of PHD inhibitors to increase hemoglobin levels can also be useful in the context of cancer by treating ischemic damage, modulating cellular metabolism, increasing levels of EPO, and increasing sensitivity to chemo- and radiotherapy [187].

A variety of assay methods have been used to detect or screen for 2-OG oxygenases, including immuno-detection of demethylated substrate peptides and formaldehyde detection in the case of KDMs. Two publicly available high-throughput screens for inhibitors of demethylases from the JMJD2/KDM4 subfamily have been reported (PubChem assay IDs 2421 (JMJD2E) and 2123 (JMJD2C)) [188]. Hydroxamic acid derivatives such as FDA-approved Vorinostat/SAHA are also histone demethylase inhibitors, with reported selectivity for the JMJD2 over PHD1/PHD2 [185]. SAHA is specifically being used to treat patients with cutaneous T cell lymphoma based on phase II single-arm clinical trial [185]. In addition, Hamada et al. designed hydroxamate analogues that selectively inhibited JMJD2C and the growth of prostate cancer cells which could be enhanced by a combinatorial treatment with LSD1 inhibitors [189].

### 4.4. Targeting Proteins Using pVHL-Based PROTACs

Monoclonal antibodies or small molecular inhibitors are commonly used chemotherapeutic agents to treat cancer by targeting specific mutant or abnormally expressed proteins. However, most intracellular proteins lack active sites or antigens for these inhibitors or antibodies to bind with and have therefore been known as “non-druggable targets.” In recent years, PROteolysis-TArgeting Chimeras (PROTACs) has emerged to be an encouraging strategy for targeted degradation of non-druggable proteins, such as scaffold proteins and transcriptional factors, via the endogenous ubiquitin-proteasome system (UPS) [190] (Figure 3). PROTACs consist of a “warhead” which binds to the target protein and a domain that binds to a ligand that recruits an E3 ligase which subsequently degrades the protein of interest.

The first generation of peptide-based PROTACs used β-TrCP or other peptides as ligands for E3 ligase to degrade target proteins. For instance, Rodriguez-Gonzalez et al. designed PROTAC-A and PROTAC-B which contain a peptide that binds to the pVHL E3 ubiquitin ligase complex and links to either dihydroxytestesterone to target androgen receptor (AR) or estradiol to target estrogen receptor alpha (ERα). Treatment of PROTAC-A and PROTAC-B inhibited the proliferation of hormone-dependent prostate and breast cancer cells in vitro by destroying AR and ERα, respectively [191]. However, the large molecular weight and associated cellular permeability issues of these peptide-based PROTACs limited their clinical utilization [190]. As a result, scientists developed small molecule E3 ligase ligands, such as pVHL, cereblon (CRBN), mouse double minute 2 homologue (MDM2), and inhibitor of apoptosis proteins (IAPs) [190]. Several small molecule VHL-based PROTACs have been reported by many groups, including PROTACs that target BCR-ABL, ALK, RTKs, SGK3, cdc20, Smad3, SMARCA2/4, bromodomain protein (BRD) 7/9, FAK, ER, and AR as reported in another review [190]. However, PROTACs are not a silver bullet; mutations in the UPS can lead to failed degradation, and activation of PROTACs is dependent on the availability of the E3 ligase which can pose as an additional disadvantage [190]. For example, pVHL-based PROTAC cannot be used to treat most kidney cancers since majority of kidney cancers lack functional VHL protein.

The latest generation of PROTACs is rendered more controllable and specific to target tissues through conditional visible or UV light activation. For instance, Kounde et al. produced caged-PROTACs by installing a photocleavable 4,5-dimethoxy-2-nitrobenzyl (DMNB) moiety onto the pVHL ligand which would block recruitment of the E3 ligase at baseline, but with irradiation at 365 nm, this caging group could be released enabling the PROTAC to degrade target proteins—in this case BRD4 [192]. Another team linked a diethylamino coumarin (DEACM) complex with the hydroxyl group of pVHL ligand to generate a pVHL-based PROTAC against estrogen-related receptor α (ERRα) and connected a photolabile 6-nitropiperonyloxymethyl (NPOM) complex with the glutarimide nitrogen in a CRBN-based PROTAC that targets BRD4. In this way, they produced two PROTACs that can degrade ERRα and BRD4 in a light-controllable manner [193]. Important considerations such as the type of E3 ligase ligands, the “warhead”, and the different linkage between them and the target protein highlight the complexities in designing and validating PROTACs. With the field only budding in recent years, it remains to be seen how this approach could be expanded to other “non-druggable” oncoproteins.

## 5. Conclusions

In this review, we summarized the current knowledge about the hypoxia signaling, including the function of HIF transcription factors, 2-OG-dependent enzymes, and pVHL, and presented some advanced studies of these oxygen sensing regulators in gene regulation of cancer progression. In summary, the oxygen-sensing system is a complicated and elaborate system containing large molecular components, including HIF transcription factors as the central regulator of oxygen homeostasis, 2-OG-dependent enzymes as the adaptive mediator, pVHL as the modulator of ubiquitin-mediated proteolysis, and the co-factors and downstream targets as functional contributors. It should be recognized that the PHDs-HIF-pVHL axis remains the best-characterized and central signaling in the oxygen sensing pathway, although novel mechanisms continue to be illuminated. Notably, more insights on 2-OG-dependent dioxygenases and their roles in oxygen sensing and cancer development are needed. Meanwhile, the tumor suppressor pVHL plays critical roles in the modulation of cancer development, especially in kidney cancer, highlighting the importance of identification of more pVHL targets (in addition to HIFs) in cancer progression. Discovery and understanding of these essential cancer pathogenesis mechanisms are beneficial and critical for developing novel pharmacological therapeutics.

## Figures and Tables

**Figure 1 ijms-21-08162-f001:**
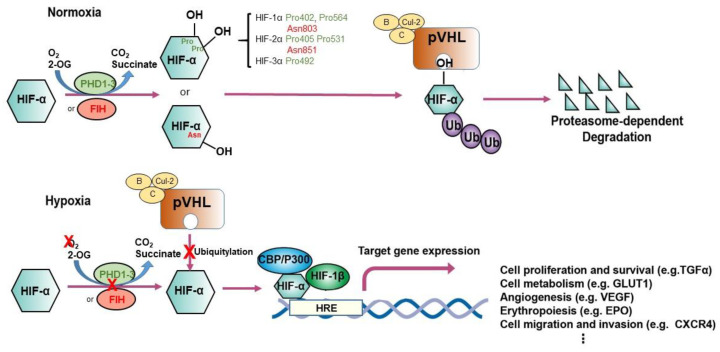
Regulation of hypoxia-inducible factor (HIF)-1α and HIF-2α. The schematic illustrates different modes of regulation of HIF-1α and HIF-2α. Under conditions of normoxia, prolyl hydroxylation regulated by EglN1-3 or asparaginyl hydroxylation regulated by factor inhibiting HIF (FIH) promote binding of HIF-α subunits to the von Hippel–Lindau (VHL) tumor suppressor protein (pVHL), the recognition component of a ubiquitin E3 ligase complex; ubiquitylation (Ub) targets HIF for proteasomal degradation. Under hypoxic conditions, the hydroxylation of HIF-α is inhibited, permitting interaction with the acetyltransferases p300 and CREB-binding protein (CBP) and further increasing transcription of HIF target genes. *EPO,* erythropoietin; *HRE*, hypoxia-responsive element, *VEGF*, vascular endothelial growth factor; *TGF-α*, transforming growth factor α; *GLUT1*, glucose transporter 1.

**Figure 2 ijms-21-08162-f002:**
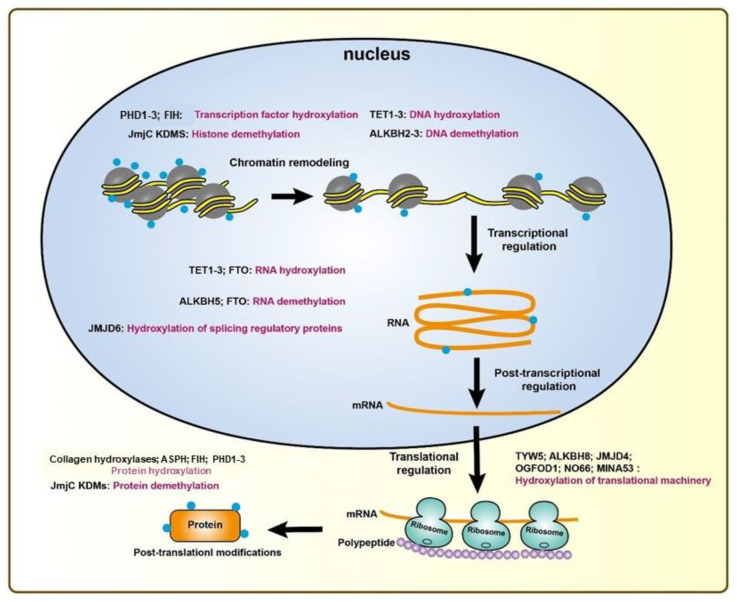
2-OG-dependent oxygenases involved in protein synthesis. 2-OG-dependent oxygenases catalyze hydroxylation and demethylation reactions that regulate transcriptional, post-transcriptional, translational, and post-translational processes [159]. *ALKBH*, alkylated DNA repair protein alkB homolog; *TET1–3*, ten-eleven translocation 1–3; *FTO*, fat mass- and obesity-associated protein; *TYW5*, tRNA wybutosine-synthesizing protein 5; *KDM*, histone lysine demethylase.

**Figure 3 ijms-21-08162-f003:**
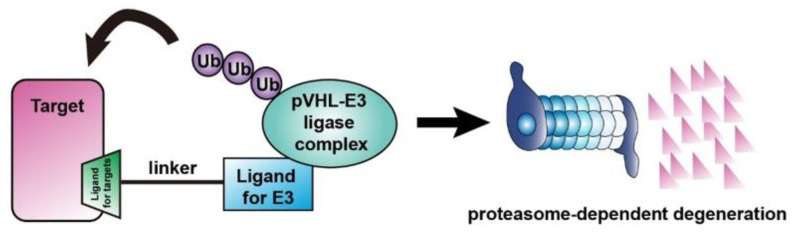
A schematic diagram for pVHL-based PROteolysis-TArgeting Chimeras (PROTACs).

**Table 1 ijms-21-08162-t001:** Compounds that have been shown to target HIF signaling.

Compound	Target	Cancer Type	Mechanism/Outcomes	Reference
Apigenin	HIF-1	Ovarian cancer	Targets PI3K/AKT signaling and downregulates HIF-1 and VEGF expression	[165,166]
SU5416	HIF-1	Acute myeloid leukemia, ovarian cancer, anaplastic thyroid carcinoma	Inhibits ability of HIF-1 to bind DNA; decreases VEGF and HIF-1α expression; downregulates VEGF, PI3K, AKT, and p70S6K1	[167,168,169]
PX-478	HIF-1	Prostate, breast, colon, and pancreatic cancer	Modulates HIF-1α deubiquitylation, inhibits HIF-1α transcription and translation	[170]
BAY 87-2243	HIF-1, HIF-2	Non-small cell lung cancer	Prevents accumulation of HIF-1α and HIF-2α	[171]
Moracin O and moracin P	HIF-1	Hepatocellular carcinoma	Inhibits HIF-1 activation	[172]
YC-1	HIF-1	Prostate cancer	Activates soluble guanylyl cyclase, induces HIF-1 degradation, targets PI3K/Akt/mTOR and NF-κB signaling	[173]
Cardamonin	HIF-1	Triple-negative breast cancer	Inhibits the mTOR/p70S6K pathway and HIF-1α expression	[174]
PT-2399	HIF-2	Clear cell renal cell carcinoma	Inhibits HIF-2α-ARNT dimerization	[175,177,178]
PT-2385	HIF-2	Clear cell renal cell carcinoma, lung cancer cell xenograft, hepatocellular carcinoma	Reduces the levels of tumor-induced circulating VEGFA, represses HIF-2α-related Stat-3/Akt/Erk signaling	[175,177,178,180,181]
TC-S 7009	HIF-2	Hepatoma	Prevents DNA binding, heterodimerization, and the transcription activation of HIF-2α by inducing allosteric conformational changes in PAS-B β-sheets	[177,182]
Compound-76	HIF-2	Breast cancer	Inhibits HIF-2α translation by promoting interaction of IRP1 and HIF-2α mRNA, represses chemoresistance and stem-ness	[183,184]
Vorinostat/SAHA	JMJD2	Cutaneous T cell lymphoma	Inhibits JMJD2	[185]

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
