# Peer review of "Hypoxia and Oxygen-Sensing Signaling in Gene Regulation and Cancer Progression"

_ijms, 2020, doi:10.3390/ijms21218162_

Round 1
Reviewer 1 Report
It would be good if the authors could explain how Hif molecules sense the molecular Oxygen (the protein that could bind to molecular oxygen) under hypoxia and normoxia
“Uncontrolled cancer cell proliferation may initially discontinuous surrounding vasculature” this statement is misleading because cancer progression eventually increases vasculature and therefore antiangiogenic therapy is one being practiced in clinic.
Please describe the acronyms as they first appear in the text. For example: pVHL
Figure is not clear how the same EglN1 and Hif were able to promote or inhibit prolyl hydroxylation. This figure could be modified for more clarity
A figure on “VHL: The proteolytic modulator of Hifs” may be helpful
“HIf-1a inhibitors section” could be more focus on direct hif-1 a targets rather indirect expression, because Hif1a can be inhibited at protein levels. For example, SU5416 is potential VEHGR2 inhibitor that may indirectly target HIf signaling. PHD inhibitors that could modulate HIf1A signaling could be included
Author Response
Response to REVIEWER 1 comments
It would be good if the authors could explain how Hif molecules sense the molecular Oxygen (the protein that could bind to molecular oxygen) under hypoxia and normoxia
Edits have been made in lines 90-93 elaborating on the mechanism of HIF oxygen sensing under hypoxia and normoxia.
“Uncontrolled cancer cell proliferation may initially discontinuous surrounding vasculature” this statement is misleading because cancer progression eventually increases vasculature and therefore antiangiogenic therapy is one being practiced in clinic.
Sentence has been replaced in lines 53-55 with a more clarifying sentence on tumor growth under hypoxia and vascularization.
Please describe the acronyms as they first appear in the text. For example: pVHL
pVHL has been defined in the abstract and first paragraph of the introduction. Revisions have been made for other acronyms as well.
Figure is not clear how the same EglN1 and Hif were able to promote or inhibit prolyl hydroxylation. This figure could be modified for more clarity
Hypoxia and normoxia effects on PHD activity and HIF hydroxylation was more clearly illustrated in the revised figures.
A figure on “VHL: The proteolytic modulator of Hifs” may be helpful
Authors believed an additional figure would be repetitive, so more detail in Figure 1 on pVHL and its function was included.
“HIf-1a inhibitors section” could be more focus on direct hif-1 a targets rather indirect expression, because Hif1a can be inhibited at protein levels. For example, SU5416 is potential VEHGR2 inhibitor that may indirectly target HIf signaling. PHD inhibitors that could modulate HIf1A signaling could be included
Lines 375-379 better introduce the therapeutic promise of targeting HIF signaling and mention PHD inhibitors as a means of modulating HIF signaling.
Reviewer 2 Report
This review manuscript summarizes accumulated knowledge of the cellular hypoxia-response system, encompassing from molecular basis to cancer pathology. I have only minor concerns, but a lot of points as below.
- Abbreviations should be defined only at the first appearance. “pVHL”, “2-OG”, “HIFs”, and “TBK1” are repetitively defined in the manuscript.
- “pVHL” and “VHL” are used mixedly. For example, “VHL” must be “pVHL” in Line 169.
- “PHD and Egln” are used mixedly in Figure 1, 2, and Page 6.
- “a-ketoglutarate” should be “2-OG” in Page 1, 8, and Figure 1.
- “proline hydroxylase (Page 1,2, and 4)” must be “prolyl hydroxylase”.
- Figure 1 should be improved. Which amino-acid residues, Pro or Asn, does the central “OH” belong to? Does pVHL-E3 recognize the HIF-a domain not related to hydroxylation? What is the beige box on the double-strand DNA? Is only the central “OH” suppressed by Egln1-3 (this should be “PHD1-3 as the text) and FIH under hypoxic conditions?
- “α-ketoglutarate dioxygenase (Page 1)” should be “α-ketoglutarate-dependent dioxygenase” or “2-OG-dependent dioxygenase”.
- Is “tumor microenvironment (TEM) (Page 2)” correct? Is it TME?
- “C-terminal transaction domain (Page 2)” should be “C-terminal transactivation domain”.
- “TFG-a (Legend for Fig. 1)” must be “TGF-a”.
- “Thus, even when both HIF-α isoforms bind to a particular control sequence, it is possible that only one is transcriptionally active (Page 3)” is obscure and should be rephrased clearly. One HIF-binding site (HRE) can be bound by only one HIF.
- In Page 4, the authors should describe that there are multiple splicing isoforms in HIF3A-gene products.
- “SU5416 blocks the interaction of VEGF and HIF-1 (Page 10)” is incorrect. SU5416 may block VEGF and HIF-1a expression.
- “1. Inhibitors for HIF-1a (Line 402)” should be “4.2. Inhibitors for HIF-2a)”.
- Table 1 indicates that the target of BAY 87-2243 is HIF-1 only, but it prevents accumulation of both HIF-1a and HIF-2a.
Author Response
Response to REVIEWER 2 comments
Abbreviations should be defined only at the first appearance. “pVHL”, “2-OG”, “HIFs”, and “TBK1” are repetitively defined in the manuscript.
Changes have been made in lines 61, 89, 208 for instance regarding the abbreviations.
“pVHL” and “VHL” are used mixedly. For example, “VHL” must be “pVHL” in Line 169.
Changes in lines 164, 166, 194, 195, and 210 for instance, regarding the usage of pVHL.
“PHD and Egln” are used mixedly in Figure 1, 2, and Page 6.
Egln1-3 has been changed to PHD1-3 in figures. EglN has been changed to PHD in line 234.
“a-ketoglutarate” should be “2-OG” in Page 1, 8, and Figure 1.
Changes have been made in figure 1 and lines 36 and 340.
“proline hydroxylase (Page 1,2, and 4)” must be “prolyl hydroxylase”.
Egln has been changed to PHD in the figures and line 234. a-KG changed to 2-OG in line 36 and in figure 1. Proline hydroxylase changed to prolyl hydroxylase in lines 14, 18, 69, 71, and 170.
Figure 1 should be improved. Which amino-acid residues, Pro or Asn, does the central “OH” belong to? Does pVHL-E3 recognize the HIF-a domain not related to hydroxylation? What is the beige box on the double-strand DNA? Is only the central “OH” suppressed by Egln1-3 (this should be “PHD1-3 as the text) and FIH under hypoxic conditions?
Amino acids on HIF-a have been more clearly labeled. VHL has been modified to include concavity to recognize HIF ODDD. Beige box has been removed and HRE region inserted. Egln1-3 has been changed to PHD1-3. Distinctions for PHD activity under hypoxia and normoxia have been made.
“α-ketoglutarate dioxygenase (Page 1)” should be “α-ketoglutarate-dependent dioxygenase” or “2-OG-dependent dioxygenase”.
Suggested changes have been made on lines 14 and 36.
Is “tumor microenvironment (TEM) (Page 2)” correct? Is it TME?
Changes have been made on lines 46 and 49 and abbreviations.
“C-terminal transaction domain (Page 2)” should be “C-terminal transactivation domain”.
Change has been made in line 89.
“TFG-a (Legend for Fig. 1)” must be “TGF-a”.
Change has been made in line 106.
“Thus, even when both HIF-α isoforms bind to a particular control sequence, it is possible that only one is transcriptionally active (Page 3)” is obscure and should be rephrased clearly. One HIF-binding site (HRE) can be bound by only one HIF.
Lines 121-124 have been rephrased to address this comment.
In Page 4, the authors should describe that there are multiple splicing isoforms in HIF3A-gene products.
Notes have been included on splicing variants of HIF-3A in lines 126-129.
“SU5416 blocks the interaction of VEGF and HIF-1 (Page 10)” is incorrect. SU5416 may block VEGF and HIF-1a expression.
Changes have been made in line 395 and table for SU5416.
“1. Inhibitors for HIF-1a (Line 402)” should be “4.2. Inhibitors for HIF-2a)”.
Change has been made in line 409.
Table 1 indicates that the target of BAY 87-2243 is HIF-1 only, but it prevents accumulation of both HIF-1a and HIF-2a.
Change has been made in the table for BAY87-2243.